# Ss4368: Pathogen-Associated Molecular Pattern for Inducing Plant Cell Death and Resistance to *Phytophthora capsici*

**DOI:** 10.3390/ijms25168674

**Published:** 2024-08-08

**Authors:** Ziwen He, Shufang Peng, Qingqing Yin, Yuanyuan Huang, Ting Deng, Yiwei Luo, Ningjia He

**Affiliations:** State Key Laboratory of Resource Insects, Southwest University, Chongqing 400715, China; ziwenhe222@163.com (Z.H.); P22977996@163.com (S.P.); yinqq516@163.com (Q.Y.); huang826yy@163.com (Y.H.); dengting0203@163.com (T.D.); luoyiwei12@swu.edu.cn (Y.L.)

**Keywords:** cell death, *Nicotiana benthamiana*, *Scleromitrula shiraiana*, plant immune inducer, pathogen-associated molecular pattern (PAMP)

## Abstract

Plant recognition of pathogen-associated molecular patterns (PAMPs) is pivotal in triggering immune responses, highlighting their potential as inducers of plant immunity. However, the number of PAMPs identified and applied in such contexts remains limited. In this study, we characterize a novel PAMP, designated Ss4368, which is derived from *Scleromitrula shiraiana*. Ss4368 is specifically distributed among a few fungal genera, including *Botrytis*, *Monilinia*, and *Botryotinia*. The transient expression of Ss4368 elicits cell death in a range of plant species. The signaling peptides, three conserved motifs, and cysteine residues (C46, C88, C112, C130, and C148) within Ss4368 are crucial for inducing robust cell death. Additionally, these signaling peptides are essential for the protein’s localization to the apoplast. The cell death induced by Ss4368 and its homologous protein, Bc4368, is independent of the SUPPRESSOR OF BIR1-1 (SOBIR1), BRI1-ASSOCIATED KINASE-1 (BAK1), and salicylic acid (SA) pathways. Furthermore, the immune responses triggered by Ss4368 and Bc4368 significantly enhance the resistance of *Nicotiana benthamiana* to *Phytophthora capsici*. Therefore, we propose that Ss4368, as a novel PAMP, holds the potential for developing strategies to enhance plant resistance against *P. capsici*.

## 1. Introduction

Filamentous fungi and oomycetes are among the most devastating plant pathogens in agricultural production, resulting in substantial annual crop losses [1]. Through long-term interaction and co-evolution, plants have evolved a dual-tiered immune response to counteract pathogen infections: pattern-triggered immunity (PTI) and effector-triggered immunity (ETI) [2]. The plant apoplast is the primary interface where initial interactions between plants and pathogens occur [3]. Within this extracellular compartment, the plant immune system detects conserved molecular patterns of pathogens via pattern recognition receptors (PRRs) localized on the cell surface, thereby activating pathogen- or microbe-associated molecular pattern (PAMP/MAMP)-triggered immunity (PTI) [4]. Upon activation, PTI elicits multiple immune responses, including the generation of reactive oxygen species (ROS), the accumulation of callose, and the upregulation of defense-related genes, which collectively serve to combat the invasion of most pathogens [2].

PAMPs are evolutionarily conserved molecules that are crucial for the fitness and survival of pathogens, and they are broadly distributed across fungi, bacteria, and oomycetes [5,6]. Classical bacterial PAMPs include flagellin, elongation factor-Tu (EF-Tu), cold shock proteins (CSPs), peptidoglycan, and lipopolysaccharides [7,8,9,10,11]. In oomycetes, known PAMPs encompass glucans, transglutaminases Pep-13, cellulose-binding elicitor lectin (CBEL), and elicitin [12,13,14,15]. Compared to oomycetes and bacteria, fungi exhibit relatively fewer identified PAMPs, such as chitin, xylanase EIX, and serine–threonine-rich glycosyl–phosphatidyl–inositol-anchored protein (SGP1) [16,17,18]. Additionally, certain PAMPs, like VmE02, are prevalent in both oomycetes and fungi [19], while necrosis and ethylene-inducing peptide 1 (Nep1)-like proteins (NLPs) and glycoside hydrolase 12 protein XEG1 are extensively distributed across oomycetes, fungi, and bacteria [20,21,22,23]. Despite the identification of numerous critical PAMPs, the diversity and underlying mechanisms of these molecules in microorganisms, particularly in fungi, remain poorly understood.

Biological factors that activate plant immunity are collectively referred to as plant immune inducers [24]. Given that PAMPs can trigger the PTI immune response, thereby enhancing resistance against a variety of pathogens, they hold significant potential as plant immune inducers. Compared to traditional chemical fungicides, which can pollute the environment and often lead to resistance development, plant immune inducers offer the advantages of being environmentally friendly, providing broad-spectrum resistance, and being highly efficient. Consequently, they have emerged as a new focal point in the field of plant protection [24,25,26]. Recently, several plant immune-inducing proteins have been successfully developed and employed as plant immune inducers. Harpin, initially characterized as a bacterial hypersensitive response-elicitor from *Erwinia amylovora*, has been shown to enhance disease resistance in a diverse range of plants and promote plant growth [27,28,29,30]. PeaT1, an elicitor identified from *Alternaria tenuissima*, has been demonstrated to enhance resistance to Tobacco Mosaic Virus (TMV) in *Nicotiana benthamiana* plants and has been developed into a commercially available protein biopesticide under the name ATaiLing [25,31]. Therefore, identifying additional plant immune inducers and exploring their potential for developing new biopesticides represent a promising future direction.

Unlike mammals, which utilize both intracellular and surface-localized PRRs to detect PAMPs, all plant PRRs are localized exclusively to the cell surface [32]. In plants, these receptors are primarily categorized into two types: receptor-like kinases (RLKs) and receptor-like proteins (RLPs). Structurally, RLKs consist of a ligand-binding ectodomain, a single-pass transmembrane domain, and an intracellular kinase domain. RLPs, while similar to RLKs, differ in that they lack a distinct intracellular kinase domain. Functionally, RLKs can serve as primary receptors, co-receptors, or regulatory proteins, whereas RLPs generally need to interact with RLKs to initiate signaling [33]. The leucine-rich repeat (LRR) RLK proteins BRI1-ASSOCIATED KINASE-1 (BAK1) and SUPPRESSOR OF BIR1-1 (SOBIR1) commonly serve as co-receptors, forming complexes with PRRs and playing a crucial role in the immune signaling process triggered by the recognition of PAMPs. For instance, the binding of XEG1 to RXEG1 (LRR) facilitates its association with the co-receptor BAK1, thereby triggering RXEG1-mediated immune responses in *N*. *benthamiana* [34,35]. Similarly, the RLP protein RE02 mediates the recognition of VmE02, a PAMP previously identified in *Valsa mali*, by forming complexes with the RLK proteins BAK1 and SOBIR1 [36]. Additionally, some apoplastic effectors induce cell death independently of BAK1 and SOBIR1, including ribonuclease Fg12 and Fg02685 [37,38].

Mulberry (*Morus* spp.) plays a crucial role in the sericulture industry, serving as the primary food source for silkworms. The fruit of the mulberry is not only delicious but also rich in vitamin C, minerals, dietary fiber, and antioxidants, among other nutrients and bioactive compounds. These health-promoting attributes have contributed to its growing popularity [39,40]. *Scleromitrula shiraiana*, the pathogen responsible for mulberry sclerotial disease, is a necrotrophic fungus that results in mummified white fruits. This disease causes severe yield losses annually in Eastern Asia [41,42]. Effectors play a key role in plant–pathogen interactions. However, the function of effector proteins in *S. shiraiana* remains unexplored. In a previous study, we identified two such proteins, sshi00003413 and sshi00010565 (now referred to as Ss4368), in *S. shiraiana* that are capable of inducing cell death [41]. Nonetheless, the precise mechanisms and biological functions underlying this cell death induction are yet to be elucidated.

In this study, we discovered that Ss4368 from *S. shiraiana* functions as a PAMP that induces cell death and immune response, thereby enhancing disease resistance against *Phytophthora capsici*. Consequently, the induced cell death is independent of BAK1 and SOBIR1 and SA pathways. In conclusion, Ss4368 shows potential to be developed as a plant immune inducer, specifically for controlling *P. capsici* infections.

## 2. Results

### 2.1. Conservation of Ss4368 Protein Sequence and Structure across Fungal Species

Ss4368 encodes a 160-amino acid protein with an open reading frame (ORF) consisting of two exons and one intron. The protein contains six cysteine residues but lacks known conserved domains (Figure 1A). Using the NCBI BLAST tool, we identified 42 homologous proteins primarily distributed in the genera *Botrytis*, *Monilinia*, and *Botryotinia*, with a significant presence noted in *Botrytis*. Phylogenetic tree analysis revealed that Ss4368 has a close evolutionary relationship with homologous proteins in *Ciborinia*, *Botrytis*, and *Botryotinia* (Figure 1B). These results suggest that Ss4368 is specifically distributed among a few fungal genera.

Multiple sequence alignment between Ss4368 and 42 homologous proteins from various fungal species revealed that all but one contained six conserved cysteine residues. The exception was a homologous protein with a mutation at the third cysteine site (Appendix A). Additionally, sequence alignment between Ss4368 and 16 homologous proteins from various races of *S. shiraiana* showed only one amino acid difference (Appendix A). MEME online analysis identified three conserved motifs in Ss4368 (Appendix A). AlphaFold online server-based predictions of the structure of Ss4368 and its homologous proteins, BCIN_02g04350 (*Botrytis cinerea* B05.10, now referred to as Bc4368), EYC84_003917 (*Monilinia fructicola*), BOTNAR_0004g00200 (*Botryotinia narcissicola*), BPAE_0117g00060 (*Botrytis paeoniae*), EAE99_005324 (*Botrytis elliptica*), and BTUL_0074g00290 (*Botrytis tulipae*), revealed that they were composed of 4–5 α-helices (Appendix A). Structural alignment using PyMOL indicated that these α-helices were highly conserved. These results suggest that Ss4368 is a highly conserved sequence and structure among these homologous proteins.

### 2.2. Ss4368 and Bc4368 Exhibit the Capacity to Induce Cell Death

To verify the activity of Ss4368 in inducing cell death across various plants, we conducted agroinfiltration experiments using pepper, tomato, and *Arabidopsis*, inoculated with *Agrobacterium* carrying the pGR107-*Ss4368* vector. The results showed that Ss4368 elicited cell death in all tested plants (Appendix A). Given the extensive distribution of Ss4368 within the *Botrytis* genus and the prominence of *B. cinerea* as a major plant pathogen in this group, we further investigated the conservation of Ss4368′s cell death-inducing activity in *B. cinerea*. We cloned the homologous gene *Bc4368* from *B. cinerea* and expressed it in *N. benthamiana* along with pGR107-*Ss4368* and *INF1*, which served as a positive control. The induction of cell death by Ss4368, Bc4368, and INF1 was determined using the electrolyte leakage method (Figure 1C,D). Semi-quantitative RT-PCR was employed to detect transcription levels (Figure 1E). The result revealed that both Ss4368 and Bc4368 induced cell death activity comparable to INF1, thereby demonstrating the high conservation of this activity in *B. cinerea*.

### 2.3. The Role of Ss4368 Signal Peptide in Protein Secretion, Localization, and Cell Death in N. benthamiana

The SignalP analysis identified a 21-amino acid signaling peptide at the N-terminal of Ss4368, suggesting that it may be a secreted protein. To verify the secretion function of this signal peptide, we performed a yeast signal sequence trap assay, which confirmed its activity. The Ss4368 signal peptide was constructed on the pSUC2 vector and transformed into YTK12 yeast. Both YTK12 carrying the Ss4368 signal peptide and the positive control Avr1b exhibited normal growth on CMD-W and YPRAA media, whereas the negative control Mg87 only grew on CMD-W medium. Additionally, enzyme activity assays revealed that the secreted invertase in YTK12 with the Ss4368 signal peptide and the positive control Avr1b effectively converted 2,3,5-triphenyl tetrazolium chloride (TTC) to insoluble red-colored 1,3,5-triphenyl formazan (TPF) (Figure 2A). These results suggest that the signal peptide of Ss4368 possesses a secretory function.

To elucidate the role of the signal peptide in the cell death activity induced by Ss4368, we generated a variant lacking this region (Ss4368^ΔSP^) and observed a delayed cell death response compared to full-length Ss4368 (Appendix A). Furthermore, the signal peptide of Ss4368 was substituted with that from *Arabidopsis thaliana* pathogenesis-related protein 1 (AtPR1). The results indicated that at 4 days post-agroinfiltration (dpa), PR1^SP^-Ss4368^ΔSP^ still induced strong cell death activity, whereas Ss4368^ΔSP^ did not (Figure 2B,C). Semi-quantitative RT-PCR was employed to detect the transcription levels of *Ss4368*, *PR1^SP^-Ss4368^ΔSP^*, *Ss4368^ΔSP^*, and *INF1* (Figure 2D). These results suggest that Ss4368 may target the extracellular space to induce intense cell death.

To further verify the localization of Ss4368 in the apoplast, we conducted subcellular localization studies on both full-length Ss4368 and Ss4368^ΔSP^ in *N. benthamiana*. The results revealed that following plasmolysis via agroinfiltration, only the fluorescence signal of Ss4368 was detected in the apoplast, whereas Ss4368^ΔSP^ was not observed (Figure 2E). These results indicate that the Ss4368 signal peptide is essential for its apoplastic localization and the induction of sufficient cell death, thereby confirming that Ss4368 functions as an apoplastic elicitor.

### 2.4. The Three Conserved Motifs of Ss4368 are Essential for Its Induction of Cell Death

Since Ss4368 lacks any known conserved domains, three conserved motifs were identified through MEME analysis. To further screen and identify the minimal region required for Ss4368-induced cell death, a truncated protein was constructed by sequentially truncating Ss4368 from the N-terminal and individually deleting each of the three motifs. Transient expression of these truncated proteins mediated by *Agrobacterium* revealed that only Ss4368-m3 could induce cell death, whereas the truncated proteins Ss4368-m1, m2, m4, m5, m6, m7, and m8 could not (Figure 3A,B). This result suggests that the three conserved motifs are essential for Ss4368 to induce sufficient cell death.

### 2.5. Mutational Analysis of Cysteine Residues in Ss4368 Reveals Critical Disulfide Bonds for Cell Death Induction

Cysteine residues play a critical role in maintaining the structural stability of proteins [43,44]. Sequence analysis revealed that Ss4368 contained six conserved cysteine residues. Further protein structure prediction indicated that Ss4368 might form three pairs of disulfide bonds: C46 with C130, C88 with C112, and C148 with C119 (Figure 3C). To investigate the effect of cysteine residues on Ss4368-induced cell death, six cysteine residues of Ss4368 were individually mutated to alanine. Upon infiltration of these mutant proteins into *N. benthamiana*, it was observed that only the C119 mutation did not affect the cell death-inducing activity of Ss4368. The C46, C130, and C148 mutations induced spotty cell death, whereas the C88 and C112 mutations completely lost this ability (Figure 3D). Transcription levels were evaluated using semi-quantitative RT-PCR (Appendix A). These results suggest that the possible disulfide bonds between C46 and C130 and between C88 and C112 are essential for Ss4368 to induce sufficient cell death, particularly the disulfide bond between C88 with C112.

### 2.6. Ss4368-Induce Immune Responses in *N. benthamiana*

To determine whether cell death induced by Ss4368 could activate plant immunity, Ss4368 was transiently expressed in *N. benthamiana*. A subsequent assessment of reactive oxygen species (ROS) burst and callose deposition was conducted using 3,3′-diaminobenzidine and aniline blue staining, respectively. The findings indicate that Ss4368 significantly enhanced the burst of ROS and the deposition of callose compared to the control (Figure 4A). Furthermore, the expressions of salicylic acid (SA), jasmonic acid (JA), and hypersensitive response (HR)-related genes were quantitatively detected from 24 to 48 h following transient expression. The results indicated that, compared to the empty vector (EV), Ss4368 significantly upregulated the expression of *NbPAL8*, *NbPAL10*, *NbPR1a*, and *NbPR2* (biosynthesis- and marker-associated gene of SA signaling), *NbAOS* and *NbLOX2* (biosynthesis-associated gene of JA signaling), and *NbHIN1* and *NbHSR203J* (marker-associated genes of HR signaling) (Figure 4B). These results indicate that Ss4368 can simultaneously activate the expression of SA, JA, and HR signaling pathways to activate the plant immune system. Furthermore, we investigated whether the plant immunity activated by Ss4368 could enhance resistance to pathogens. Since *S. shiraiana* was not pathogenic to *N. benthamiana*, two fungal pathogens, *Sclerotinia sclerotiorum* and *B. cinerea*, which are closely related to *S. shiraiana*, were selected for inoculation experiments. Additionally, *P. capsici*, an oomycete, was also chosen for these experiments. The results demonstrated that the overexpression of Ss4368 significantly enhanced resistance to *P. capsici*, but not to *Sclerot. sclerotiorum* and *B. cinerea* (Figure 5A–C). Additionally, the homologous protein Bc4368 also enhanced resistance to *P. capsici* in N. benthamiana (Figure 5D). Taken together, these results suggest that Ss4368 can activate ROS burst, callose deposition, and SA, JA, and HR signaling pathways, thereby enhancing resistance against *P. capsici*, an oomycete.

### 2.7. Ss4368/Bc4368-Mediated Cell Death: Independent from BAK1, SOBIR1, and Salicylic Acid Signaling Pathways

Cell death and immune responses induced by PAMPs are typically triggered by the recognition of cell surface pattern recognition receptors, with BAK1 and SOBIR1 serving as two major co-receptors involved in PRRs-mediated ligand recognition and signal transduction [19,45]. Consequently, we investigated whether the cell death induced by Ss4368 was dependent on BAK1 and SOBIR1. Homozygous mutants of *bak1* and *sobir1*, created using CRISPR/Cas9 knockout techniques, were employed in these experiments [34,35,46]. Ss4368, Bc4368, and INF1 were transiently expressed in wild-type (WT) and mutant plants. The results indicated that Ss4368, Bc4368, and INF1 induced cell death in WT plants. Additionally, Ss4368 and Bc4368 were capable of inducing cell death in *bak1* and *sobir1* mutants, whereas INF1 was not (Figure 6A). These results suggest that the cell death triggered by Ss4368 and Bc4368 occurs independently of BAK1 and SOBIR1. In addition, Ss4368, Bc4368, and INF1 were expressed in *NahG* gene overexpression plants to assess whether impaired SA synthesis could disrupt Ss4368/Bc4368-induced cell death. The results demonstrated that Ss4368, Bc4368, and INF1 induced cell death in NahG plants, suggesting that the SA signaling pathway is not essential for Ss4368/Bc4368-induced cell death. The extent of cell death from Ss4368, Bc4368, INF1, and buffer treatments was quantified with electrolyte leakage assays. Transcription levels were evaluated using semi-quantitative RT-PCR (Figure 6B).

## 3. Discussion

During interactions between plants and pathogens, PAMPs secreted by pathogens are recognized by PRRs on the surface of plant cells. This recognition triggers a PTI response, which generally serves to inhibit the invasion of a wide array of pathogens, highlighting the potential of PAMPs as agents for enhancing plant immunity. Despite this potential, the identification and application of PAMPs have been limited. In this study, a novel PAMP, Ss4368, was identified. This PAMP is specifically distributed among a few fungal genera, including *Botrytis*, *Monilinia*, and *Botryotinia*. The plant immunity induced by Ss4368 and the homologous protein Bc4368 enhances resistance to *P. capsici*. Additionally, the cell death triggered by Ss4368 and Bc4368 occurs independently of the conventional co-receptors BAK1, SOBIR1, and SA signaling pathways.

### 3.1. Ss4368 Represents a Specific Elicitor Found in Fungi

Only 42 homologous proteins to Ss4368 were identified using BLAST in the NCBI database. These homologous proteins are predominantly specific to fungal genera such as *Botrytis*, *Monilinia*, and *Botryotinia*, suggesting that Ss4368 is evolutionarily specific and restricted to a limited number of fungal genera. This is in contrast to other elicitors’ homologs, such as XEG1, identified in *P. sojae*, which is prevalent across fungi, oomycetes, and bacteria [23]; VmE02 in *V. mali* is widespread among fungi and oomycetes [19]; and SGP1 in *Ustilaginoidea virens* is extensively distributed within fungi [18]. Notably, a significant number of homologous proteins are present in *Botrytis,* indicating a high conservation of Ss4368 within this genus. Despite the limited distribution of Ss4368 among fungi, its homologous proteins are highly conserved both structurally and in terms of cell-death-inducing activity. This conservation suggests that Ss4368 plays an important role in the interactions between these specific fungal genera and their host plants. The ability of Ss4368 to induce cell death responses in a diverse array of plants, similar to those reported for VmE02 and XEG1, suggests that the recognition receptors responsive to Ss4368 in these plants are broadly conserved.

Disulfide bonds are crucial for the function and structure of proteins. For instance, mutations in the cysteine residues C38 and C44 of the effector protein SsSSVP1^∆SP^ disrupt its ability to induce cell death, form homo-dimers, and interact with the target protein QCE8 [47]. Similarly, mutations in the cysteine residues C59 and C75 of the apoplastic effector Pep1 impair its pathogenicity in *Ustilago maydis* [48]. Structural predictions suggest that Ss4368 may form three pairs of intracellular disulfide bonds, with point mutation experiments revealing that C46 and C130, as well as C88 and C112, are essential for the induction of sufficient cell death, particularly C88 and C112. Interestingly, the C119 mutation did not impact Ss4368-induced cell death activity, while the C148 mutation significantly compromised it. This discrepancy may be attributed to the presence of two additional cysteine residues on the alpha helix hosting C119, which may compensate for the mutation and maintain overall protein stability. In contrast, the alpha helix containing C148 possesses only a single cysteine residue, and its mutation adversely affects the structure of Ss4368. The observation of a mutation in the amino acid corresponding to position C119 in the homolog of Ss4368 suggests that the cysteine residue at this position may not be critical for its function or structure. This finding adds to the understanding of the structural requirements for Ss4368-induced cell death. The ability of the Ss4368, Ss4368^ΔSP^, and PR1^SP^-Ss4368^ΔSP^ to induce cell death and Ss4368, Ss4368^ΔSP^ subcellular localizations were investigated. These results demonstrated that apoplastic localization is essential for Ss4368 to effectively induce cell death, aligning with the known properties of other PAMPs such as XEG1 [23], VmE02 [19], and SGP1 [18]. Notably, XEG1 and SGP1 entirely lost their cell death-inducing capabilities post de-signaling peptide, whereas VmE02 and SsCP1 [49] exhibited delayed cell death activity, similar to that of Ss4368 when de-signaled. This suggests that there may be intracellular receptors in *N. benthamiana* that recognize Ss4368, VmE02, and SsCP1; however, the immune response triggered by these intracellular receptors is slower compared to that of pattern recognition receptors. The truncation of the three motifs conserved in Ss4368 revealed that these motifs are essential for the induction of cell death by Ss4368. This suggests that the conserved motifs are necessary for the recognition of Ss4368 by unidentified receptors.

### 3.2. Ss4368 Is a Novel PAMP Elicitor in Inducing Resistance to Hemibiotrophic *P. capsici*

Ss4368 activates plant immunity by inducing ROS burst, callose deposition, and the expression of genes associated with JA, SA, and HR. These results confirm that Ss4368 functions as an elicitor, capable of triggering a comprehensive defense mechanism in plants. Inoculation experiments further demonstrated that Ss4368 enhances resistance to *P. capsici* (hemibiotrophs) while having no discernible impact on resistance to necrotrophic pathogens, such as *Sclerot. sclerotiorum* and *B. cinerea*. VmE02 from *V. mali* enhances resistance to *P. capsici* and *Sclerot. sclerotiorum* by activating plant immunity in *N. benthamiana* [19]. XEG1 from *P. sojae* enhances resistance to *P. parasitica var nicotianae* by activating plant immunity in *N. benthamiana* [23]. These results suggest that, although different PAMPs can activate plant immunity, their efficacy against various pathogens differs in *N. benthamiana*. Additionally, the resistance conferred by Ss4368 is particularly effective against hemibiotrophic pathogens, which exploit living plant tissue initially before transitioning to dead tissue. The study hypothesizes that Ss4368 may also exhibit resistance to biotrophic pathogens, although this requires further experimental verification. The differential impact of Ss4368 on necrotrophic pathogens leads to the proposal of two potential mechanisms. Firstly, necrotrophic pathogens may suppress Ss4368-mediated plant immunity through the secretion of alternative effector proteins. Secondly, while the initial immune response triggered by Ss4368 may suppress infection, the cell death induced by Ss4368 could paradoxically promote necrotrophic pathogen infection, creating a dynamic balance between resistance and susceptibility.

Hemibiotrophic pathogens, including *P. capsici,* are prevalent in agriculture and cause significant damage to major crops [50,51]. Traditional methods of disease control, such as breeding for resistance and the use of chemical agents, have limitations. In contrast, plant immune inducers offer advantages such as shorter development times, lower costs, and reduced environmental impact [25]. The Harpin protein from *E. amylovora* is a notable example of a plant immune inducer, known for its ability to induce disease resistance through salicylic acid-mediated systemic acquired resistance [28]. Given Ss4368′s capacity to stimulate multiple immune responses and enhance resistance to *P. capsici*, it holds significant potential for development as an immune inducer against this pathogen. The specificity of Ss4368, found only in a few fungal genera, could minimize immune escape by these hemibiotrophic pathogens, prevent resistance failure, and extend the durability of the resistance.

### 3.3. Cell Death Induced by Ss4368 Does Not Depend on Conventional Co-Receptors

Plants typically employ PRRs located on their cell membranes to recognize PAMPs in the extracellular space, thereby triggering pattern-triggered immunity (PTI) responses [32]. PRRs are structurally categorized into two classes: RLP and RLK. Among RLKs, BAK1 and SOBIR1 commonly act as co-receptors, forming complexes with PRRs to facilitate intracellular signal transduction upon PAMP recognition [45]. Previous research has shown that PAMPs like the glycoside hydrolase family 12 protein FoEG1 from *Fusarium oxysporum* and VmE02 from *V. mali* induce cell death and immune responses that are dependent on the co-receptors BAK1 and SOBIR1 [19,36,52]. However, other PAMPs, such as SGP1 from *U. virens* and XEG1 from *P. sojae*, trigger cell death and immune responses solely through BAK1, without the involvement of SOBIR1 [18,23]. In contrast, certain secreted effectors can induce cell death and immune responses independently of both BAK1 and SOBIR1, as exemplified by the ribonuclease Fg12 from *F. graminearum* [37].

In this study, it was determined that Ss4368, an apoplastic elicitor, induces cell death independently of the co-receptors BAK1 and SOBIR1 on the cellular surface. This finding suggests that while all these effectors function as PAMPs, they induce plant immunity through distinct mechanisms. This result implies that plants may recognize Ss4368 through a RLP and subsequently recruit additional RLKs to transmit downstream signals. Alternatively, plants may directly engage other RLKs to activate immunity without the need for BAK1 and SOBIR1.

## 4. Materials and Method

### 4.1. Plant and Strain Culture Conditions

*S. shiraiana* (SX-001), *Sclerot. sclerotiorum* (BB-1), *P. capsici,* and *B. cinerea* (B05.10) were cultured on potato dextrose agar (PDA) medium at 24 °C. The isolate of *S. shiraiana* (SX-001) was collected from mulberry sclerotial disease fruit in Chongqing, China, in August 2019. The isolate of *Sclerot. sclerotiorum* (BB-1) was collected from oilseed rape sclerotium in Chongqing, China, in April 2022. Prof. Wenxing Liang provided the *B. cinerea* B05.10 strain, and Prof. Guanhua Ma provided the *P. capsici* strain. *Escherichia coli* strains *Trans*1-T1 (Transgen, Beijing, China) were cultured on Luria–Bertani (LB) medium at 37 °C. The *Agrobacterium tumefaciens* GV3101 and GV3101 (pJIC SA_Rep) (Weidi, Shanghai, China) were cultured on LB medium at 28 °C. *N. benthamiana* plants and *bak1*/*sobir1*-knockout mutants were cultivated in a climate chamber at 24 °C with a 16 h light/16 h dark cycle and 70% humidity.

### 4.2. Bioinformatic Analysis

The SignalP 5.0 server (https://services.healthtech.dtu.dk/service.php?SignalP-5.0, accessed on 18 January 2024) was utilized to predict the presence of signal peptides [53]. Using the blastP search function (https://blast.ncbi.nlm.nih.gov/Blast.cgi, accessed on 10 March 2024) with default settings in the NCBI database, homologous proteins of Ss4368 were identified across various pathogen species. The ClustalW program was employed for multiple sequence alignments of Ss4368 and homologous proteins.

The phylogenetic tree of Ss4368 and its homologous proteins was constructed using the maximum likelihood method and the Poisson model in MEGA 11 software [54], and subsequently refined for visual presentation using iTOL v6 online software. Conserved motifs in Ss4368 and its homologous proteins were identified using the MEME suite (https://meme-suite.org/meme/tools/meme, accessed on 13 March 2024) [55]. The protein structure of Ss4368 was predicted using AlphaFold 2 [56], whereas the structures of homologous Ss4368 proteins were retrieved from the UniProt database (https://www.uniprot.org/, accessed on 5 June 2024).

### 4.3. Plasmid Construction

The cDNAs from *S. shiraiana* (SX-001) were used as a PCR template to amplify the coding sequences of the *Ss4368* genes using 2 × Super Pfx Master Mix Polymerase (Cwbio, Taizhou, China). The homologous gene of *Ss4368* in *B. cinerea* was synthesized by Sangon Biotech, Co. (Shanghai, China). These genes were cloned into vectors based on homologous recombination technology using the *pEASY*-Basic Seamless Cloning and Assembly kit (Transgen, Beijing, China). For *Agrobacterium*-mediated transient expression in *N. benthamiana*, gene fragments of *Ss4368* and *Ss4368* mutants were separately cloned into the PVX vector pGR107 digested with specific enzymes (*Bsu15*Ⅰ(*Cla*Ⅰ) and *Sal*Ⅰfor pGR107). Using the Fast Mutagenesis System kit (Transgen, Beijing, China), six cysteine residue mutants of Ss4368 were created, adhering to the manufacturer’s guidelines. To confirm the efficacy of the Ss4368 signal peptides, a signal peptide of *Ss4368* was cloned into the pSUC2 vector digested with *EcoR*Ⅰ and *Xho*Ⅰ enzymes. To create constructs for confocal microscopy, gene fragments of *Ss4368* were ligated to the pBinGFP4 vector digested with *Kpn*Ⅰ and *BamH*Ⅰ enzymes, which contain a C-terminal GFP tag. To create constructs for inoculation assay, gene fragments of *Ss4368* were ligated to the pBinGlyRed3 vector digested with *Xba*Ⅰ and *Xho*Ⅰ enzymes. The primers employed in this study are detailed in Appendix A.

### 4.4. A. tumefaciens Agroinfiltration

The constructs were chemically transformed into *A. tumefaciens* strain GV3101. Monoclones that tested positive in PCR assays were cultured in an LB medium supplemented with antibiotics. The cultures were incubated at 28 °C in a shaking incubator at 200 rpm overnight. The cultures were harvested by centrifugation and then resuspended in infiltration buffer (10 mM MgCl_2_, 10 mM MES, 200 μM acetosyringone) to adjust the optical density (OD_600_) to 0.8. After incubation in a dark room at room temperature for 2–3 h, the abaxial surface of 4-week-old *N. benthamiana* leaves was infiltrated using a needle-free syringe.

### 4.5. Electrolyte Leakage Measurement

Electrolyte leakage was assessed using a method adapted from previously established protocols [57]. Disks measuring 8 mm in diameter were taken from *N. benthamiana* leaves and submerged in 5 mL of distilled water at room temperature for 3 h. The initial electrical conductivity (EC1) was measured using an SX-650 conductivity meter (Sanxin, Shanghai, China). Subsequently, the leaf disks were boiled for 25 min in the solution within sealed tubes. After cooling to room temperature, the final electrical conductivity (EC2) was measured. The percentage of electrolyte leakage was calculated using the following equation: electrolyte leakage (%) = (EC1/EC2) × 100.

### 4.6. Confocal Microscopy

*Ss4368/Ss4368^ΔSP^*-GFP and *AtPIP2A*-mCherry were co-expressed in *N. benthamiana* via *Agrobacterium*-mediated transformation. For confocal microscopy, leaves were harvested 36–48 h post-infection and imaged with an FV1200 confocal laser scanning microscope (Olympus, Tokyo, Japan). To induce plasmolysis, *N. benthamiana* leaves were soaked in 1 M NaCl solution on glass slides for 5–10 min. The AtPIP2A protein from *A. thaliana* served as a plasma membrane marker. The GFP was excited using an argon laser at 488 nm, with emission detection spanning 500 to 550 nm. The mCherry fluorescence was excited using a 559 nm laser, allowing for the detection of emissions in the range of 600 to 680 nm.

### 4.7. Analysis of Signal Peptide Function in Yeast

Studies investigating the function of signaling peptides in yeast utilize protocols established by previous research [58]. The signal peptide of Ss4368 was amplified with specific primers, cloned into the pSUC2 vector, and fused with the invertase gene. This construct, pSUC2-*Ss4368^SP^*, was then transformed into the YTK12 yeast strain, with positive clones being identified through screening on the CMD-W medium. Yeast strains exhibiting invertase enzymatic activity demonstrated normal growth on YPRAA medium and facilitated the reduction of 2,3,5-Triphenyltetrazolium Chloride (TTC) to insoluble, red-colored 1,3,5-Triphenylformazan (TPF). YTK12 yeast transformed with the pSUC2-*Avr1b^SP^* vector served as the positive control, while those transformed with the empty pSUC2 or pSUC2-*Mg87^N^* vector were used as negative controls.

### 4.8. Inoculation Assay

The pBinGlyRed3 (serving as a control) or pBinGlyRed3-*Ss4368* constructs were introduced into the leaves of 4-week-old *N. benthamiana* via *A. tumefaciens* at an optical density of 0.8 (OD_600_). Subsequently, 24 h after infiltration, regions of the leaves were inoculated with 10 mm diameter mycelial plugs of *Sclerot. sclerotiorum* (strain BB-1, 2-day-old colony), *P. capsici* (4-day-old colony), or *B. cinerea* (strain B05.10, 3-day-old colony). After inoculation, the leaves were incubated at 24 °C under high humidity and darkness. Lesion diameters were measured, and images were captured under ultraviolet (UV) light 1–1.5 days post-inoculation.

### 4.9. RNA Isolation and Quantitative Reverse Transcription-PCR (qRT-PCR) Analysis

To extract RNA, plant tissue was placed in a 1.5 mL tube containing two metal beads and immediately submerged in liquid nitrogen. The samples were then homogenized using an MM 400 Mixer Mill (Retsch, Haan, Germany) for 1 min at 30 Hz. Total RNA was isolated using TRIzol reagent (Invitrogen, Carlsbad, CA, USA) according to the manufacturer’s instructions. The RNA concentrations were quantified using a NanoDrop spectrophotometer (Thermo Fisher, Woburn, MA, USA). Reverse transcription of approximately 1000 ng of RNA was performed using PrimeScript™ RT reagent (Takara, Dalian, China). qRT-PCR assays were carried out using Super Real PreMix Plus SYBR Green (Tiangen, Beijing, China) on the Step One Plus Real-Time PCR System (Applied Biosystems, Waltham, MA, USA). The *NbEF1a* was used as internal controls for *N. benthamiana*. All primers used in the qRT-PCR assays are listed in Appendix A.

### 4.10. ROS and Callose Staining

The control or pBinGlyRed3-*Ss4368* construct was introduced into *N. benthamiana* leaves, which were then allowed to infiltrate for 36 h before undergoing ROS and callose staining. For ROS staining, the leaves were treated with 1 mg/mL 3,3-Diaminobenzidine (DAB) for 10 h, followed by decolorization with ethanol. For callose staining, the leaves were similarly decolorized with ethanol and then stained overnight with 0.01% aniline blue in darkness. Images were captured using a microscope (Olympus, Tokyo, Japan).

### 4.11. Statistical Analysis

Statistical analysis was carried out using GraphPad Prism version 9.3.1 (GraphPad, San Diego, CA, USA). Differences between two distinct samples were assessed using a two-tailed unpaired Student’s *t*-test, and *p*-values were calculated accordingly. The results are expressed as mean ± standard deviation (SD).

### 4.12. Accession Numbers

The nucleotide and protein sequences utilized in this study were sourced from the NCBI. The accession numbers are as follows: Ss4368 (WYV95643.1), Bc4368 (XP_001552893.1), EYC84_003917 (KAA8574667.1), BPAE_0117g00060 (TGO23911.1), EAE99_005324 (KAF7927947.1), BTUL_0074g00290 (TGO13222.1), and BOTNAR_0004g00200 (TGO70035.1).

## 5. Conclusions

This study has successfully identified Ss4368 as a novel PAMP with a unique distribution among fungal species. Our findings demonstrate that the cell death activity induced by Ss4368 and its homologous proteins operates independently of BAK1, SOBIR1, and SA signaling pathways. This independence from conventional immune signaling pathways highlights a distinct mechanism by which Ss4368 can elicit plant defense responses. Moreover, the immune response triggered by Ss4368 specifically enhances the resistance in *N. benthamiana* against *P. capsici*, indicating that this PAMP has the potential to be developed as a targeted plant immune inducer. The implications of these findings not only expand our understanding of plant immune recognition and signaling but also pave the way for the development of new strategies to combat plant diseases. Additionally, future structural studies of Ss4368 and its plant receptor using crystallography or nuclear magnetic resonance (NMR) could provide deeper insights into plant–pathogen interactions and lay the groundwork for molecular breeding efforts aimed at increasing disease resistance in plants.

## Figures and Tables

**Figure 1 ijms-25-08674-f001:**
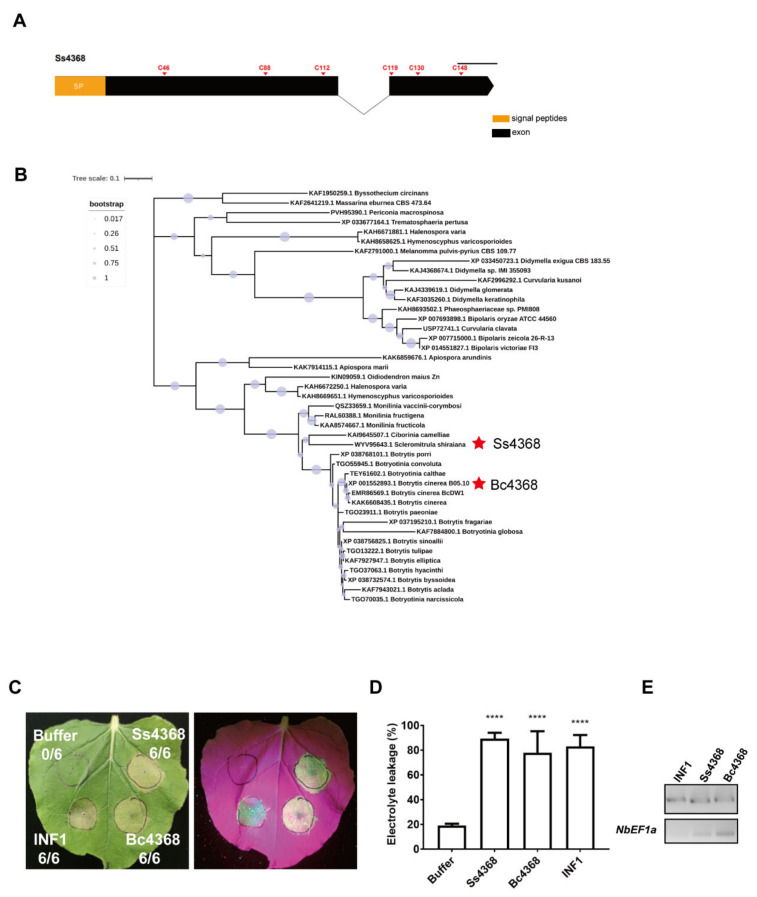
Distribution of Ss4368 homologous protein in fungi and their induction of cell death in *Nicotiana benthamiana*. (**A**) Diagrammatic representation of the Ss4368 coding sequence (CDS). The signal peptide is highlighted in orange, with the two exons depicted in black, and the six cysteine residues are labeled. (**B**) Phylogenetic analysis of Ss4368 and its 42 homologous proteins, derived from various fungi, was conducted using the maximum likelihood algorithm. Bootstrap values for each branch are denoted by the size of the violet circles, ranging from 0.017 to 1. Ss4368 and its homologous protein, Bc4368, are highlighted with asterisks. (**C**) Induction of cell death in *N. benthamiana* by Ss4368 and Bc4368. Leaves of *N. benthamiana* were infiltrated with *Agrobacterium tumefaciens* carrying plasmids pGR107-*Ss4368* or pGR107-*Bc4368*. INF1 and buffer served as positive and negative controls, respectively. Symptoms of cell death in *N. benthamiana* were photographed under white and UV light 4 days post-agroinfiltration. The figures represent the ratio of leaves displaying cell death to the total leaves evaluated. (**D**) Quantification of cell death by measuring electrolyte leakage. The values represent the mean ± standard deviation (SD) based on three biological replicates. Significant differences compared to controls (buffer) were identified using Student’s *t*-test (**** *p* < 0.0001). (**E**) Semi-quantitative RT-PCR analysis of the expression levels of *Ss4368*, *Bc4368*, and *INF1* genes transiently expressed in *N. benthamiana*. *NbEF1a* serves as the internal reference gene.

**Figure 2 ijms-25-08674-f002:**
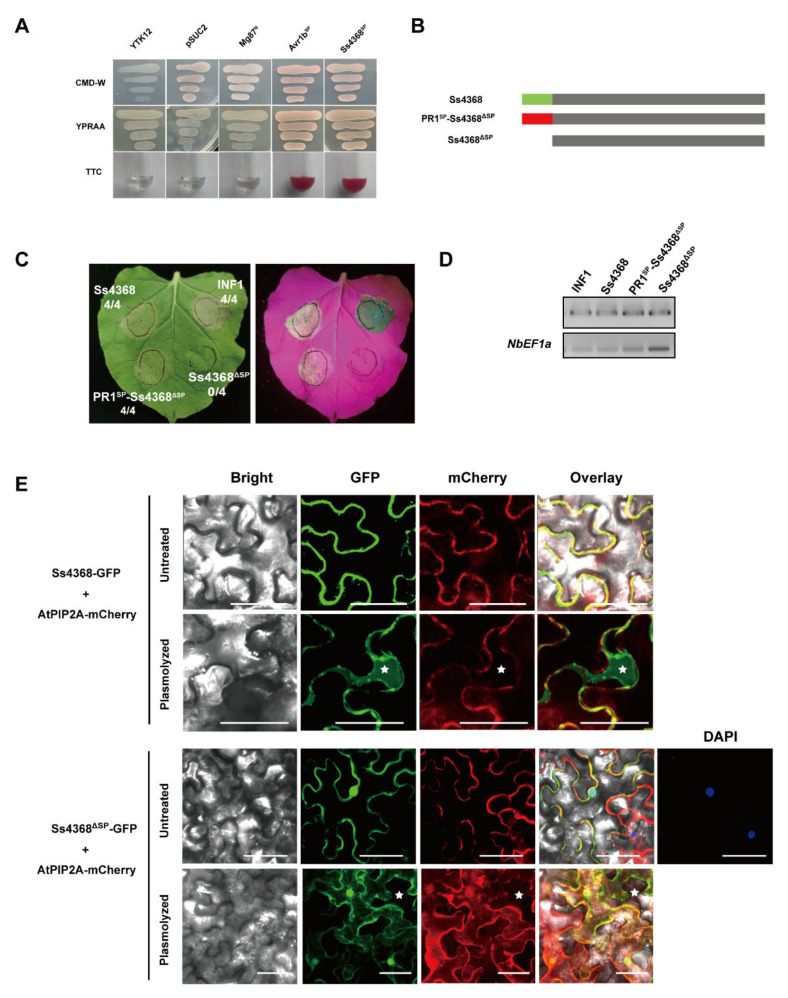
Essential role of the Ss4368 signaling peptide in apoplastic localization and induction of cell death. (**A**) The functionality of the Ss4368 signal peptide was confirmed using a yeast invertase secretion assay. CMD-W medium was used to verify the presence of the pSUC2 vector in YTK12. YPRAA medium and a color reaction were used to assess the secretory function of the signal peptide. Negative controls included the YTK12 strain with and without the pSUC2 vector, as well as YTK12 carrying pSUC2-*Mg87^N^* (the N-terminal sequence of Mg87). The positive control was YTK12 carrying pSUC2-*Avr1b^SP^* (the signal peptide of Avr1b). (**B**) Schematic representation of the constructs for Ss4368, Ss4368 without signal peptide (Ss4368^ΔSP^), and the fusion mutant PR1^SP^- Ss4368^ΔSP^. (**C**) Cell death phenotypes were observed in *Nicotiana benthamiana* leaves transiently expressing Ss4368, Ss4368^ΔSP^, and PR1^SP^-Ss4368^ΔSP^. Images were captured under white and UV light 4 days post-agroinfiltration. The figures represent the ratio of leaves exhibiting cell death to the total leaves evaluated. (**D**) Semi-quantitative RT-PCR analysis of the expression levels of *Ss4368*, *Ss4368^ΔSP^*, *PR1^SP^-Ss4368^ΔSP^*, and *INF1* genes transiently introduced into *N. benthamiana*. *NbEF1a* was used as the internal reference gene. (**E**) Subcellular localization of Ss4368 and Ss4368^ΔSP^ in *N. benthamiana*. *Agrobacterium* strain GV3101, harboring the fusion constructs *Ss4368*-GFP and *Ss4368^ΔSP^*-GFP, was used for transient expression in *N. benthamiana* epidermal cells. Fluorescence images were captured 2 days post-agroinfiltration. Pre- and post-plasmolysis images are shown in the upper and lower panels, respectively. Plasmolysis of *N. benthamiana* leaves was induced by immersion in 1 M NaCl solution for 5–10 min. The gene encoding *Arabidopsis thaliana* plasma membrane intrinsic protein 2A (AtPIP2A) was tagged with mCherry to serve as a membrane localization marker. DAPI was utilized as a nuclear marker, and asterisks indicate the apoplastic space. Scale bars measure 50 μm.

**Figure 3 ijms-25-08674-f003:**
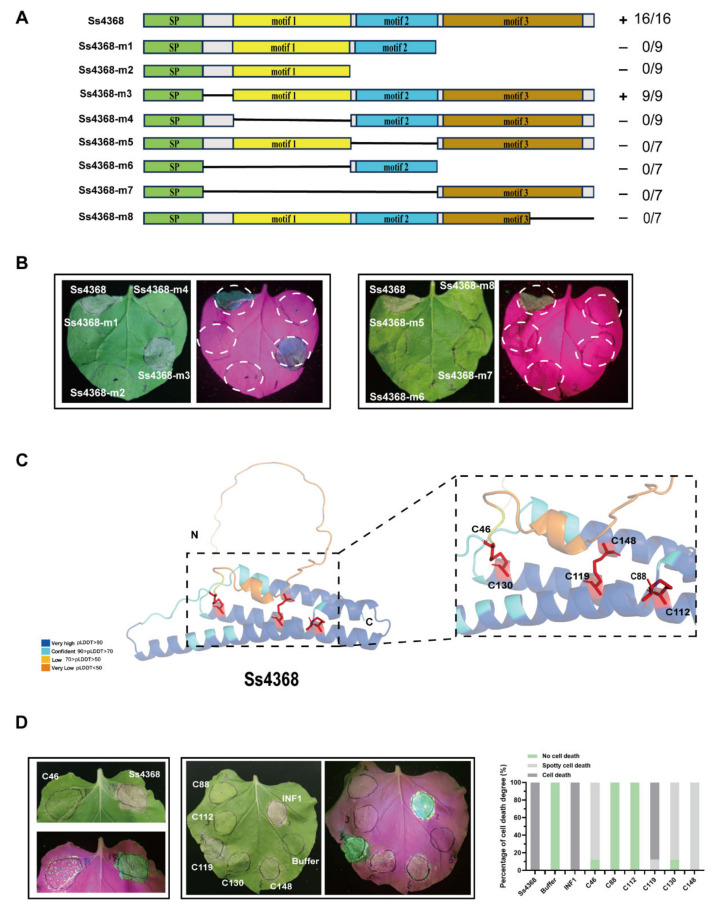
Functional verification of conserved motifs and disulfide bonds in Ss4368. (**A**) Schematic representation of Ss4368 protein truncations. *Agrobacterium tumefaciens* carrying pGR107-*Ss4368* and its truncated variants were infiltrated into *Nicotiana benthamiana* leaves. The symbol “+” indicates a protein that induces cell death, whereas “-” signifies a protein that does not. The figures represent the ratio of leaves displaying cell death to the total leaves evaluated. (**B**) Images depicting the characteristic symptoms under white and UV light at 4.5 days post-transient expression of Ss4368 and its truncated variants. (**C**) The structural conformation prediction of the Ss4368 protein using the AlphaFold 2 algorithm. The confidence score for each residue (pLDDT) is color-coded, with red bars highlighting the predicted potential of three pairs of disulfide bonds. The enlarged image on the right displays three potential disulfide bond pairs: C46 with C130, C119 with C148, and C88 with C112. (**D**) Observation of cell death phenotypes in *N. benthamiana* leaves expressing Ss4368 and its cysteine residue variants. *Agrobacterium tumefaciens* carrying pGR107-*Ss4368* and its cysteine residue variants were infiltrated into *N. benthamiana* leaves. Cell death symptoms in *N. benthamiana* were documented using white and UV light photography 7 days after agroinfiltration. The degree of cell death induced by Ss4368 was quantitatively assessed. Cell death was categorized into three levels: no cell death, spotty cell death, and cell death, across a sample of 8 leaves.

**Figure 4 ijms-25-08674-f004:**
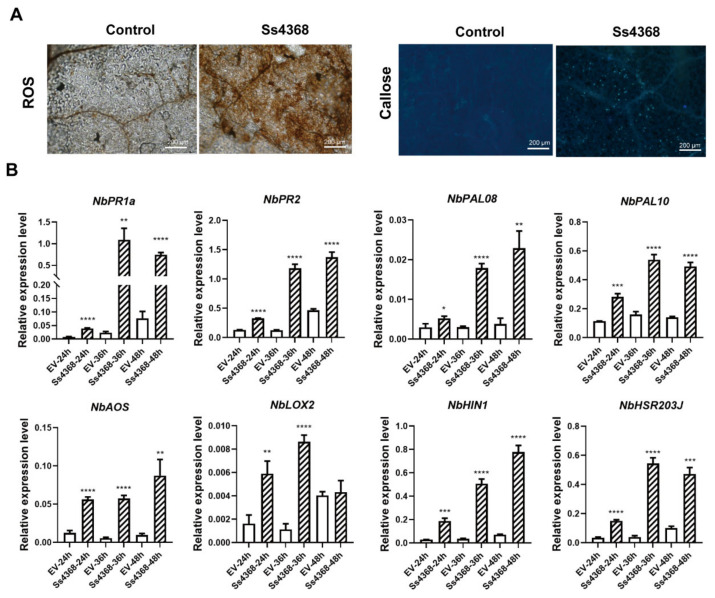
Induction of plant immunity in *Nicotiana benthamiana* by Ss4386. (**A**) ROS accumulation and callose deposition in *N. benthamiana* expressing Ss4368. Leaves of *N. benthamiana* were infiltrated with a buffer control and with Ss4368 constructs for 2 days. Post infiltration, the leaves were stained with 3,3′-diaminobenzidine and aniline blue and photographed using a fluorescent microscope. Scale bar: 200 μm. (**B**) Quantification of transcriptional levels of genes associated with salicylic acid (SA), jasmonic acid (JA), and the hypersensitive response (HR) via quantitative RT-PCR (qRT-PCR) after the transient expression of an empty vector (EV, serving as a control) and Ss4368 at 24, 36, and 48 h. *NbEF1a* served as the housekeeping gene. Values represent mean ± SD (*n* = 3 replicates). Significant differences compared to controls were determined using Student’s *t*-test (* *p* < 0.05, ** *p* < 0.01, *** *p* < 0.001, **** *p* < 0.0001).

**Figure 5 ijms-25-08674-f005:**
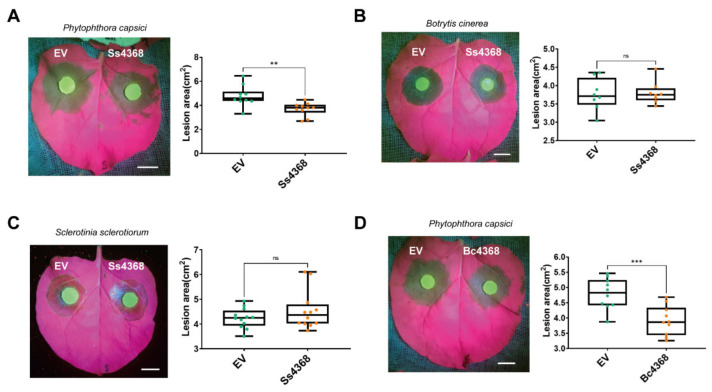
Enhancement of *Phytophthora capsici* resistance in *Nicotiana benthamiana* by Ss4368 and Bc4368. (**A**–**C**). Leaves of *N. benthamiana* were separately infiltrated with an empty vector (EV, as a control) and Ss4368 for 24 h. Subsequently, these leaves were inoculated with mycelial plugs from *P. capsici* (**A**), *Botrytis cinerea* (**B**), and *Sclerotinia sclerotiorum* (**C**). Representative photographs were captured 1.5 d (**A**,**B**) or 1 d (**C**) post-inoculation (dpi). The lesion area was recorded under UV light to assess the disease severity. Mean ± SD values are shown (*n* > 7 biological replicates). (**D**) Leaves of *N. benthamiana* were separately infiltrated with EV and Bc4368 for 24 h before inoculation with *P. capsici*. Representative images were taken after 1.5 dpi. The lesion area was recorded under UV light. Mean ± SD values are shown (*n* = 10 biological replicates). Significant differences compared to controls were determined using Student’s *t*-test (ns = not statistically significant, ** *p* < 0.01, *** *p* < 0.001). Scale bar: 1 cm.

**Figure 6 ijms-25-08674-f006:**
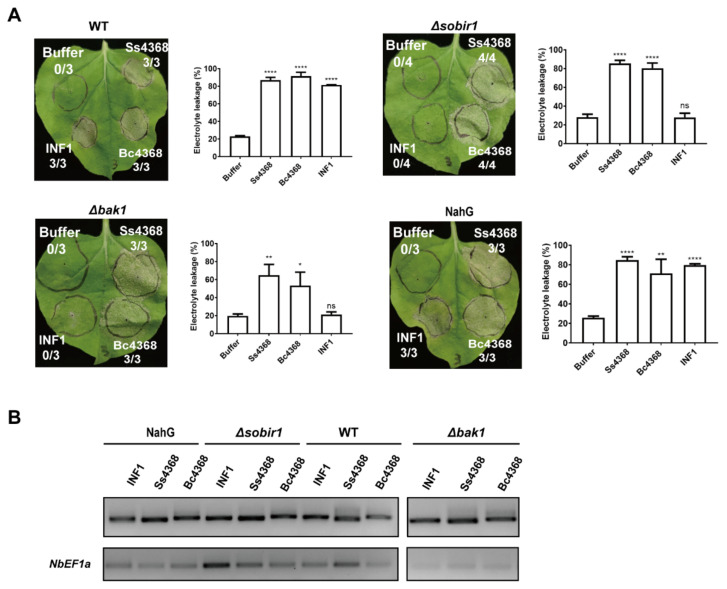
Induction of cell death in *Nicotiana benthamiana* by Ss4368 and Bc4368 occurs independently of BAK1, SOBIR1, and salicylic acid. (**A**) Cell death phenotypes in *N. benthamiana* leaves from wild-type, *NahG* gene overexpression lines, *sobir1* or *bak1* knockout mutants after infiltration with a buffer control and constructs of Ss4368, Bc4368, and INF1. *Agrobacterium tumefaciens* strains harboring pGR107-*Ss4368*, *Bc4368*, and *INF1* were used for transient expression in *N. benthamiana* leaves. INF1 and buffer were used as positive and negative controls, respectively. Photographs capturing typical symptoms were taken 4.5 days post-agroinfiltration. The figures represent the ratio of leaves displaying cell death to the total leaves evaluated. Cell death quantification was performed using electrolyte leakage measurements. Values represent mean ± SD (*n* = 3 biological replicates). Significant differences compared to controls were determined using Student’s *t*-test (ns = not statistically significant, * *p* < 0.05, ** *p* < 0.01, **** *p* < 0.0001). (**B**) Expression levels of *Ss4368*, *Bc4368*, and *INF1* were measured using semi-quantitative RT-PCR. *NbEF1a* was used as the internal reference gene.

## Data Availability

The data that support the findings of this study are available from the corresponding author upon reasonable request.

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
