# Peer review of "Ss4368: Pathogen-Associated Molecular Pattern for Inducing Plant Cell Death and Resistance to Phytophthora capsici"

_ijms, 2024, doi:10.3390/ijms25168674_

Round 1

Reviewer 1 Report

Comments and Suggestions for Authors

The manuscript is well-written and the author effectively discusses the role of pathogen-associated molecular patterns (PAMPs) in inducing plant cell death and resistance to Phytophthora capsici. However, a few more changes are suggested to enhance the overall quality of the work.

·        Please check the font size in sections 2.6 and 3.2. 

·        In a few places, the scientific names are not italicized.

·  The discussion section could be further improved by incorporating preliminary studies or similar efforts in other PAMPs across different plant species.

·        The conclusion section can be strengthened by discussing how detailed structural studies, such as crystallography or NMR, could provide a clearer picture of the 3-D structure of Ss4368 and its interactions with plant receptors, as well as potential future applications of PAMPs in enhancing plant resistance.

Comments on the Quality of English Language

Minor editing of the English language is required.

Author Response

Comments 1: Please check the font size in sections 2.6 and 3.2.

Response 1: Sorry for the mistake. The error you mentioned has been corrected in the revised manuscript at lines 251 and 388.

Comments 2: In a few places, the scientific names are not italicized.

Response 2: Thank you for pointing this out. We have revised this throughout the text.

Comments 3: The discussion section could be further improved by incorporating preliminary studies or similar efforts in other PAMPs across different plant species.

Response 3: We appreciate this valuable feedback. Consequently, we have incorporated additional research advances related to PAMPs and discussed these in conjunction with our findings on Ss4368. The newly added discussion can be found on lines 359-363 and 394-399 of the revised manuscript. Corresponding references have been included at lines 711-716.

Comments 4: The conclusion section can be strengthened by discussing how detailed structural studies, such as crystallography or NMR, could provide a clearer picture of the 3-D structure of Ss4368 and its interactions with plant receptors, as well as potential future applications of PAMPs in enhancing plant resistance.

Response 4: We agree with this comment. Therefore, we have revised this section to emphasize this point. In the conclusion, the relevant statement has been included at lines 576-580.

Comments 5: Minor editing of the English language is required.

Response 5: We apologize for the language issues present in the original manuscript. We have now worked on both language and readability and have also involved native English speakers for language corrections. We really hope that the flow and language level have been substantially improved.

Reviewer 2 Report

Comments and Suggestions for Authors

The entire manuscript has been prepared very carefully. In the Introduction all important aspects related to this work have been included. The basic elements of the Methodology are presented in Materials and Methods. On the other hand, many detailed elements concerning the methodological procedure are given in Results. This form makes it very easy to follow how the authors carried out individual experiments. The results are presented in a transparent way. As a result of well-designed experiments, the Authors achieved very valuable results, of which the following are particularly noteworthy: i) identification of a novel PAMP, Ss4368, ii) finding that Ss4368 is restricted to a limited number of fungal genera including Botrytis, Monilinia, and Botryotinia, iii) finding that the plant immunity (N. benthamiana) induced by Ss4368 enhances resistance to Pytophthora capsici, iv) finding that Ss4368 activates plant immunity by inducing ROS burst, callose deposition, and the expression of genes associated with Jasmonic acid, Salicylic acid, and HR, and v) that resistance associated with Ss4368 is particularly effective against hemibiotrophic pathogens. The discussion is substantive and very interesting. The conclusion is written synthetically. This manuscript should be published in IJMS/ MDPI. I have only a few, minor comments.

Remarks

Line 253 N. benthamiana - different font size?

Line 385 P. capsici - different font size?

Line 437-439 For all strains of fungi and bacteria, please provide: where they were isolated from (origin) and date of isolation. If their sequences are deposited in Genbank, please provide numbers

Line 437  S. shiraiana, S. sclerotiorum - please note that you cannot use the same letter "S." for two different genera of fungi, i.e. Scleromitrula and Sclerotinia

Line 463 Agrobacterium - it should be in italic

Line 518-519 ‘..of S. sclerotiorum (strain BB-1), P. capsici, or B. cinerea (strain B05.10)’ – state how old the colonies were from which the mycelial plugs were taken

In References, the names of plants and fungi should be in italic, for example:

Line 603 Phytophthora  - it should be in italic

Line 613 Ustilaginoidea virens - it should be in italic

Line 625 Arabidopsis -- it should be in italic [and other literature]

Author Response

Comments 1: Line 253 N. benthamiana - different font size?

Response 1: Sorry for the mistake and thank you for pointing this out. The corresponding revision has been made in the revised manuscript at line 251.

Comments 2: Line 385 P. capsici - different font size?

Response 2: Sorry for the mistake. The corresponding revision has been made in our revised manuscript at line 388.

Comments 3: Line 437-439 For all strains of fungi and bacteria, please provide: where they were isolated from (origin) and date of isolation. If their sequences are deposited in Genbank, please provide numbers

Response 3: We agree with this valuable comment. The origin of different fungi and bacteria has been added at lines 445-452. The sequence for Ss4368 has been submitted to Genbank and can be found in the material in 4.12 at line 562.

Comments 4: Line 437 S. shiraiana, S. sclerotiorum - please note that you cannot use the same letter "S." for two different genera of fungi, i.e. Scleromitrula and Sclerotinia

Response 4: Thank you for pointing this mistake out. To avoid confusion, we have abbreviated the Latin name Scleromitrula shiraiana to S. shiraiana and Sclerotinia sclerotiorum to Sclerot. Sclerotiorum throughout the text. The changes can be found in our revised manuscript at line 444.

Comments 5: Line 463 Agrobacterium - it should be in italic

Response 5: Sorry for the mistake. The corresponding revision has been made in our revised manuscript at line 475.

Comments 6: Line 518-519 ‘..of S. sclerotiorum (strain BB-1), P. capsici, or B. cinerea (strain B05.10)’ – state how old the colonies were from which the mycelial plugs were taken

Response 6: We agree with this valuable comment. Information on the age of S. sclerotiorum (strain BB-1), P. capsici, and B. cinerea (strain B05.10) cultures on PDA plates used for inoculation experiments has been added to our revised manuscript at lines 530-531.

Comments 7: In References, the names of plants and fungi should be in italic, for example:

Line 603 Phytophthora - it should be in italic

Line 613 Ustilaginoidea virens - it should be in italic

Line 625 Arabidopsis -- it should be in italic [and other literature]

Response 7: Sorry for the mistake and thank you for pointing this out. We have italicized all plant names and strain names in the references.